# The Effects of the Type of Information Played in Environmentally Themed Short Videos on Social Media on People’s Willingness to Protect the Environment

**DOI:** 10.3390/ijerph19159520

**Published:** 2022-08-03

**Authors:** Shiyong Zheng, Jiarong Cui, Chaojing Sun, Jiaying Li, Biqing Li, Weili Guan

**Affiliations:** 1School of Business, Guilin University of Electronic Technology, Guilin 541004, China; shiyongzheng123@whu.edu.cn (S.Z.); lijiaying226699@163.com (J.L.); zhanrui0688h@guet.edu.cn (B.L.); 2College of Digital Economics, Nanning University, Nanning 530200, China; 3School of Management, Hainan University, Haikou 570228, China; 4China Justice Big Data Institute, Beijing 100041, China; cuijiarong1107@163.com; 5Shandong Labor Vocational and Technical College, Jinan 250000, China; research_123@foxmail.com

**Keywords:** short videos, message type, environmental intention

## Abstract

This study used a 2 × 2 experimental design to explore the effects of message type (non-narrative vs. narrative information) and social media metrics (high vs. low numbers of plays) of low-carbon-themed social media short videos on people’s willingness to protect the environment. Subjects completed questionnaires after viewing short videos that contained different message types and social media metrics, and a final sample of 295 cases was included in the data analysis. The study found that, while the type of information (i.e., non-narrative or narrative) of the low-carbon-themed social media short videos had no direct effect on people’s willingness to protect the environment, its indirect effects were significant. These indirect effects were achieved through immersion experience and social influence. Subjects who watched narrative videos had a higher level of immersion experience, which in turn was significantly and positively correlated with environmental intention; meanwhile, those who watched non-narrative videos experienced a higher level of social influence, which in turn was significantly and positively correlated with environmental intention. In addition, subjects who viewed high-volume videos experienced a more positive effect on their willingness to protect the environment. This study explored how message design could promote subjects’ perceptions and positive attitudes towards environmental protection, with important managerial implications.

## 1. Introduction

In September 2020, China announced to the world at the United Nations General Assembly the goal of achieving carbon peaking by 2030 and carbon neutrality by 2060 [1]. In response to environmental protection issues, although China has formulated laws and regulations to conserve resources, and has also promoted scientific guidance, the reality of its environmental problems has not yet been effectively addressed [2,3]. The general public lacks awareness of environmental protection, and simply understands it as sanitation, greening, etc. [4]. People believe that environmental protection is the responsibility of the government, and do not take into account the impact of their own activities on the environment [5]. Thus, the participation of citizens in environmental protection is low, and a culture of universal participation in environmental protection is far from being established [6].

In the age of mobile internet, social media has become an important tool for people to learn, work, and live [7]. Compared to traditional methods of environmental protection publicity, the use of online media to organize and carry out publicity work could further improve the efficiency and quality of establishing environmental awareness [8]. However, the effectiveness of environmental protection propaganda on social media is currently not satisfactory [9]; effective means of improving people’s environmental behaviors through short social media videos is a matter of concern.

Social media is a powerful tool for public education, and learning about environmental protection through it could be an effective way to increase the public’s willingness to protect the environment [10]. Such knowledge about environmental protection and the importance of lowering carbon emissions should be continuously disseminated and promoted on social media platforms [11]. Based on these circumstances, this study concerned itself with how environmental protection messages on social media influenced the public’s willingness to protect the environment, and how these messages could be designed to promote positive attitudes towards proactive environmental protection [12].

Within the framework of immersion theory and social influence theory, this study examined the effects of different types of messages (non-narrative vs. narrative messages) and social media indicators (low vs. high video views) in influencing the public’s awareness and behavior toward environmental protection [7,13]. In addition, this study also explored the mechanisms underlying the roles of non-narrative messages versus narrative messages as well as the influences of social media indicators.

Specifically, this study had three objectives. The first objective was to examine which type of video (non-narrative or narrative) was more effective at influencing public attitudes toward voluntary environmental protection. The second objective was to examine which indicator was more effective at influencing public willingness to protect the environment, high-volume vs. low-volume short video plays. The third objective was to explain the mechanisms underlying different video types (non-narrative or narrative), in order to understand which was more effective at influencing public attitudes toward voluntary environmental protection. As such, this study examined the mediating role of immersive experiences and social influence, i.e., immersive experience and social influence mediating the impact of video types (non-narrative or narrative) and the impact of social media indicators (low vs. high video views) on people’s environmental intentions.

In order to answer the above questions, the authors used an experimental approach and designed four experimental videos (i.e., non-narrative vs. low play, non-narrative vs. high play, narrative vs. low play, and narrative vs. high play) in which subjects filled out questionnaires after watching the videos to answer questions about their immersion experience, social influence, and environmental intentions.

## 2. Theory and Hypothesis

### 2.1. Types of Messages

Although existing research has demonstrated the positive impact of media messages on individuals’ attitudes and behavioral intentions toward environmental protection, the question of which message type is more persuasive in addition to what its mechanisms of action are needed to be further explored [14]. While traditional environmental messages have used didactic and explanatory approaches to educate and persuade audiences, in recent years the use of narrative forms of messaging has become an effective means of persuasion [15]. As such, this approach to messaging has received widespread attention from researchers [16]. A narrative is a series of coherent stories with a clear beginning, passage, and ending [3]. It contains information about scenes, characters, and conflicts; the story presents unresolved or ongoing problems, and offers solutions [17]. Narrative messages have a special persuasive power, as well-designed storylines and recognizable characters can capture an audience’s attention and stimulate personal, emotional responses [18]. These, in turn, can influence the audience’s beliefs, attitudes, and willingness to act [19]. In contrast to narrative messages, non-narrative messages, a traditional type of persuasive message, rely on rhetorical argumentation and the presentation of factual information to support a claim, primarily through explanation, illustration, education, and advocacy non-narrative messages persuade by communicating the message clearly to the audience [20]. Non-narrative messages tend to aggregate multiple cases, providing more general content through quantitative descriptions, and providing conclusions that can be broadly applied to the group as a whole [21]. Both narrative and non-narrative messages are common means of presenting environmental information, in order to trigger a change in individual environmental attitudes [22]. However, there have been conflicting conclusions regarding the comparison of the effectiveness between narrative vs. non-narrative messages. The first view is that, overall, narrative messages are more effective when geared toward the general public, particularly when people lack experience and knowledge [23]. Exposure to narrative messages has been thought to have a more significant persuasive effect on people’s self-efficacy than non-narrative messages [24]. This finding has been validated in empirical studies where narrative messages were shown to have demonstrated a positive impact on individual attitudes, and were more effective in influencing attitudes compared to non-narrative messages [25]. The second view is that non-narrative information, particularly statistical information, is more effective. A study by Cialdini (1991) found that, overall, statistical information had a greater impact on individual risk perceptions than narrative information [21]. An empirical study found that non-narrative messages were more effective in communicating basic prevention knowledge than narrative messages during public health emergencies, and that individuals had more positive attitudes toward the effectiveness of the preventive measures presented [26]. Researchers found, in studies of the effectiveness of online science videos, that traditional explanatory videos have an advantage over entertainment formats, and are considered to be more authoritative and credible [27]. A third view is that there is not yet enough evidence to prove which is more effective between the two messaging types. For example, on the issue of people’s environmental attitudes, the results of statistical analyses have not unequivocally proven which message type is more dominant [28]. Moreover, there have been differences found in the persuasive effects of the two, making each messaging type suitable for different purposes. Studies have found that non-narrative messages are more effective than narrative messages in terms of their impact on individual attitudes; on the other hand, narrative messages are more effective in terms of their impact on behavioral intentions [29]. In the practice of science communication, traditional forms of non-narrative messages still exist, while the role of narrative messages is beginning to emerge. It is not yet known what the impact of these two approaches will be on people’s attitudes toward environmental protection [25].

In this study, we classified video genres as narrative or non-narrative [28]. In cases of viewers watching science videos (narrative vs. non-narrative), both genres promote their environmental intentions, while the differences in the two video genres do not affect viewers’ environmental intentions [21]. Therefore, we make the hypotheses that follow.

**H1.** 
*Video type does not influence viewers’ willingness to protect the environment.*


The cognitive processes of individuals in response to narrative and non-narrative information are quite different [30]. During the processing of narrative information, individuals temporarily move away from pre-existing schemas and experiences, immerse themselves in the narrative, and change their attitudes by reducing rebuttals and increasing their personal emotional engagement with the story [29]. In comparison, the cognitive process for non-narrative messages requires the audience to pay critical attention to the main ideas of the message, evaluate the message in relation to established attitudes, knowledge, experiences, and other ideas, and understand the content through logical thinking and argumentation, which in turn triggers attitude change [31]. Research has found that narrative advertising is more likely to trigger an emotional response in audiences than non-narrative advertising (argumentative advertising). This approach allows audiences to experience the same emotional state as the storyteller through plot and character, whereas non-narrative advertising is not causal and story-oriented, and is less likely to trigger immersion as well as emotional responses to the message [32]. Environmental issues are social issues that require not only personal engagement and immersion in the message, but also require a certain amount of engagement with environmental knowledge. Such issues require people to engage in cognitive responses and to think critically [25]. Based on this, it became necessary to investigate the pathways of influence of the two types of messages and, in conjunction with previous literature, examine how media messages impact individual attitudes through immersion experiences and social influence [31].

### 2.2. Immersion Experience

Researchers have defined the immersion experience to be a unique psychological process in which an individual’s attention, imagery, and sensations create a fusion, and where there is complete focus on the events occurring in the narrative [33]. Immersion is characterized by the fact that an audience in the process of being immersed may lose track of time and be unable to observe what is happening around them [34]. This loss of touch with the real world may occur on a physical level, where the individual becomes so immersed in the experience and may no longer notice the presence of other people [8,35]. More importantly, however, this immersion occurs on a psychological level; when a person is immersed in a story, they may become dull to the contradictions between the narrative story and the real world, and not notice where the two contradict each other [36].

Researchers interviewed professional and non-professional people, such as dancers, rock climbers, and surgeons, and found that they were extraordinarily enthusiastic and engaged in their work [37]. Research suggests that concentration and enjoyment are two important parts of the immersion experience. This has been confirmed in HCI research, where the immersion experience is not only about personal concentration and enjoyment, but also about exploration and entertainment [38]. For example, the immersion experience can lead to enjoyment through engagement in the activity, and consumers’ subjective experiences. Early research has found that interactive technologies lead to more immersive experiences, and that once online users are immersed, they are motivated to invest more time and effort [18]. In studies of online business environments, the mediating effects of immersive experiences have generally focused on site navigation or on the use of communication tools, such as instant messaging.

Although individuals may engage with and enjoy non-narrative information, the immersive experience itself is primarily in response to narrative information [25]. Narrative immersion experience theory explains the process by which narrative information comes into play through the immersion experience, suggesting that narrative persuasion arises when individuals become immersed in the narrative world. This creates an emotional connection with the story characters, which is unlikely to provoke viewer rebuttal; this results in changes in individuals’ beliefs and attitudes that are consistent with the narrative’s message [39]. The immersion experience distinguishes narrative information processing from non-narrative information processing, which clearly has a persuasive purpose, does not create another world for the individual, and is less likely to stimulate empathy or create mental imagery [40]. One research study found that narrative information exposure significantly predicted audience immersion levels [41], with individuals exposed to narrative information becoming more immersed in the message. Another advertising study also found that individuals who read narrative magazine advertisements produced higher levels of immersion experiences compared to those who read non-narrative advertisements (advocacy messages) [20].

This led our research to propose Hypothesis 2, which follows.

**H2.** 
*Watching narrative videos produces a higher level of immersive experience compared to non-narrative videos.*


While scholars have identified the importance of immersive experiences in commercial activities, some have argued that immersive experiences are not related to marketing. More research has confirmed the influence of immersive experiences in online environments [42]. Researchers found that immersive experiences positively influenced the behavior of internet users, including their willingness to revisit websites [3]. Research has tended to explore how to enhance the immersive experiences of users. Immersive experiences play an important role in changing attitudes and beliefs, and their role in doing so is important. In the case of violence and crime, research has found that audiences with high immersion experiences exhibit belief attitudes that are consistent with the narrative [43]. In a study on the topic of support for homosexuality, researchers found that individuals with higher levels of immersion produced attitudes that were consistent with the information they were exposed to [44]. Furthermore, immersive experiences mediated the effects of narrative information exposure on individuals’ willingness to go to bed early [45]. A similar mediating role has been validated in other studies, such as brand communication intentions [22], e-cigarette refusal attitudes, and behavioral intentions [42]. In these studies, narrative information exposure was significantly associated with higher levels of immersion experience, and the level of immersion experience was significantly and positively associated with individual attitudes. In terms of the impact of narrative information, the subject of the study was less important than engagement with the narrative itself [46]. Thus, in the context of environmental protection and the population, the authors argued that the use of immersion experience theory was applicable. Guided by immersion experience theory, this study argued that people who watch narrative videos would develop higher levels of immersive experiences, would become more immersed in the narrative world, and hence develop more positive attitudes toward environmental protection.

Based on the above discussion, we proposed Hypothesis 3.

**H3.** 
*The level of immersion is positively correlated with people’s willingness to protect the environment, i.e., the higher the level of immersion, the stronger people’s willingness to protect the environment.*


### 2.3. Social Influence

While immersion theory assumes that the process of narrative persuasion is isolated, and that individuals play an active role in interpreting narrative messages, it is worth noting that narrative persuasion also occurs within social groups, such as family, friends, professional networks, and consumer communities. Individuals are motivated to act in a similar way to others, and their perception of the behavior of the reference group will influence their own subsequent behavior [47].

Social influence refers to the fact that consumers who have already adopted a product or service will have an influential effect on consumers who have not yet adopted it [48]. In other words, when consumers are new to a product or service, they are influenced by the groups or individuals around them, and may develop a herd mentality, or may demand more information about the product. This information will influence the influenced person’s judgement in the product decision-making process, in terms of adoption, rejection, or repeat purchasing. Social influence theory states that individuals in a social network are influenced by others to change their attitudes and behaviors. This social influence can be divided into normative and informational influences [25,30].

Perceived social influence exists at the psychological level of the individual, and represents each individual’s understanding of the prevalent norms of behavior and collective norms [49]. Social influence is defined as the norms of behavior based on group identities that are transmitted and understood through social interaction. It is divided into informational and social influences.

Normative influence is the compulsion to adopt similar usage behaviors to those of other groups around them as a result of normative pressures generated by those around them. This normative pressure creates a degree of psychological conformity, as the influenced person feels punished or rejected by others if they do not adopt the same consumption behavior as the individuals around them or as the group in question [26]. In order to escape this punitive situation, the influenced person adopts behavioral traits that are consistent with others, thereby attenuating this negative psychological cue. Furthermore, it has been shown that, the higher the social status and social class of the influencer, the stronger this normative influence will be, creating a greater impact on the influenced [50].

Informational influence refers to the desire of consumers to have access to more sources of information during the process of making consumption decisions, in order to ensure that consumption decisions will be more accurate [51]. In the process of informational influence, consumers are judged less by the social status and influence of the influencer, and more by the accuracy of the information that the influencer obtains, and whether or not it provides them with an adequate amount of information on which to base their decisions. The influenced person uses the influenced individuals or groups around them as a source of information, and consumer decision-making behavior depends on the information provided by these groups [37].

Social influence builds understanding among group members through communication (Rimal and Real, 2003). Informational and normative influences are thought to be influenced by others, but fictional stories may also encourage shifts in normative beliefs [51]. Non-narrative information describes behavioral norms in the form of statistics or legal texts, which have certain universal and coercive binding effects, and can influence individuals’ perceived social influence [47].

Therefore, this study argued that the perception of social influence was primarily influenced by non-narrative information, and consequently proposed Hypothesis 4.

**H4.** 
*Viewing non-narrative videos will produce higher levels of social influence compared to narrative videos.*


Social influences further affect individuals’ behavioral intentions [52]. For example, individuals perceive norms through interactions with members of their social networks; furthermore, normative perceptions of friends’ brand attitude behavior are significantly and positively related to their own brand attitude behavioral intentions [53]. The effects of social influence on individual attitudes and behavioral intentions has been validated in studies of other behaviors related to environmental protection, such as waste sorting and new energy vehicle purchases. In a behavioral study of new energy vehicle purchases, Moran et al. (2016) found that narrative films had an indirect effect on consumer purchase intentions through social influence. In the case of environmental issues, the vast majority of consumers’ environmental intentions were influenced by social pressures from family, friends, and other sources. On topics that were not directly observable, individuals gained an accurate understanding of social conditions and responded effectively through social influence when they were not clear on what behaviors were appropriate [54].

Environmental protection has a certain threshold of knowledge for the general public; for example, the identification of recyclable waste, toxic waste, etc. in the waste sorting process. The general public does not typically have a great deal of knowledge or past experience to draw upon, hence observing the way others behave becomes an important cue for action. When individuals perceive that the majority of society approves of and adopts environmentally friendly behavior, individuals are likely to perceive this as socially acceptable, and tend to follow this perceived norm themselves; this results in a more positive attitude toward environmentally friendly behavior. Based on this information, Hypothesis 5 of this study was proposed.

**H5.** 
*Social influence is positively related to the willingness of people to protect the environment, i.e., the higher the level of social influence perceived by people, the stronger their willingness to protect the environment.*


### 2.4. Social Media Indicators

Unlike traditional media, social media can be used to indicate how the general public reacts to a video, and how popular a video is by displaying a range of quantitative indicators. The social media indicators referred to in this study are mainly interaction data that can be obtained from online activities, such as numbers of visits, clicks, readings, retweets, comments, likes, favorites, private messages, votes and recommendations, etc. [55]. The digital nature of social media metrics provides direct social cues for audiences who use these digital metrics to infer others’ perceptions of media content; consequently, their perceptions of environmental messages may in turn be influenced by these digital metrics. A content analysis of environmental blog posts found that there is a significant positive relationship between the number of comments posted by readers per week and the happiness of bloggers [56]. After manipulating two metrics (audience ratings and views of online news articles), researchers found that individuals spent more time reading articles with higher ratings. The effect exerted by view counts showed a curvilinear effect, whereby readers were more inclined to read articles that had fewer views or more views [57]. Manipulating just one metric, YouTube video views, also found that views influenced subjects’ perceptions of the importance of the global warming issue, with subjects who watched highly viewed videos being more likely to believe that global warming is an important issue for most Americans [58]. Overall, social media metrics influence audiences’ perceptions of reality; moreover, messages with high social media metrics are seen as more socially acceptable, and are more likely to influence individuals’ attitudes towards their beliefs than videos with low social media metrics [59]. With the widespread dissemination of social media videos, especially among young people, short videos are a form of environmental messaging that warrants attention.

This study sought to explore the role of social media indicators in influencing people’s willingness to protect the environment by manipulating the level of video plays. Based on the above findings, the authors hypothesized that social media videos with high airplay would have a more significant impact, and hence proposed Hypothesis 6.

**H6.** 
*Watching high-volume videos will produce a more immersive experience compared to low-volume videos.*


Social media metrics can influence the perceived social influence on individuals. Social networks allow users to express themselves and make themselves visible to their social networks. In this way, social networking sites provide users with opportunities to understand and communicate cultural norms and social cues [60]. Evidence of the impact of social media metrics on perceived social influence is provided by research on brand identity, where researchers manipulated two types of reviews of approved and disapproved brands. The authors found that reviews of approved brands had a significant effect on an individual’s social influence [61]. When social media metrics do not clearly indicate whether viewers approve of the message or not, individuals still interpret social media metrics as cues to social influence. Research on consumer brand attitudes that manipulated the numbers of views, comments, and shares of short videos found that social media metrics and perceived similarity worked together for social influence, with higher levels of perceived similarity associated with higher levels of social influence for high social media metrics [62]. This study argues that video airplay provides cues to individuals’ perceived social influence, and that high airplay affects people’s perceptions of environmental behavior as well as their degree of personal compliance, leading to our formulation of Hypothesis 7.

**H7.** 
*Watching high-volume videos produces higher levels of social influence compared to low-volume videos.*


The hypothetical model for this study is shown in Figure 1.

## 3. Research Methodology

In order to test the research hypotheses, one pre-experiment and one formal experiment were designed in this paper, which are described below.

The purpose of the pre-experiment was to select the experimental scenario manipulation materials that would be applicable to the different groups of short video types and video plays in the subsequent formal experiment [12]. The formal experiment was designed to confirm the consistent effects of changes in subjects’ environmental intentions induced by specific short video types and video playback volumes, as well as the mechanisms of the mediating variables [10].

### 3.1. Pre-Experiment

A pre-experiment was conducted prior to the formal experiment, with the purpose of selecting appropriate stimulus scenarios for the formal experiment (narrative vs. non-narrative video types, and high vs. low playback). In order to eliminate confounding variables that interfere with the results as much as possible, as well as for authenticity and credibility, we chose short environmental videos on social media as the experimental material to ensure high reliability and validity of the experimental study [11].

#### 3.1.1. Pre-Experiment 1

First, we selected two types of short environmental videos (narrative vs. non-narrative) that were based on short environmental videos from several social media platforms [63]. The narrative video depicts a fisherman whose work has been relocated to land in response to the national policy of the Yangtze River fishing ban in China, and who has been able to live a happy life with the help of the government [5]. The non-narrative video presents an explanation of China’s fishing ban on the Yangtze River, and the penalties for violating the ban [64].

We invited 60 undergraduate students (27 males and 33 females, average age 20–25 years) from a comprehensive university in Wuhan to conduct a questionnaire test on each of the above materials in order to verify the manipulation of video genres. We edited the experimental videos with the help of Python technology and the editcool tool in order to ensure that the length (3 min), sound quality, brightness, and pixels of the videos remained consistent, and also to exclude as much as possible other confounding variables [65]. We based the measure on 52 validated questionnaires, where respondents indicated to what extent that they agreed (1 = strongly disagree; 7 = strongly agree) with the type of environmental short video to which the appeal belonged; all scales for this study’s experiment were based on a 7-point Likert scale. The test results showed that narrative scores were significantly higher in narrative videos than in non-narrative videos (M_narrative_ 5.23, M_non-narrative_ = 3.39, *p* < 0.001), while narrative scores were significantly lower in non-narrative videos than in narrative videos (M_non-narrative_ = 3.17, M_narrative_ = 5.43, *p* < 0.001). Thus, the experiment validated the manipulation of video type.

#### 3.1.2. Pre-Experiment 2

Subsequently, 60 undergraduate students (35 female, 25 male, mean age = 19.8 years) were invited to participate in this experiment, which aimed to examine the validity of the manipulation of high vs. low video playback. Respondents were randomly assigned into the high vs. low playback experimental groups [66]. Students participating in the test watched the short environmental video from Pre-experiment 1, but this time they were grouped into two high and low groups based on the number of likes, retweets, comments and favorites [10]. After watching the video, students were asked to fill in a questionnaire.

Perceived video play was measured on a 7-point scale (1 = low play; 7 = high play) [62]. Our results, based on 55 validated questionnaire measures, confirmed that there was a significant difference in perceived video playback between playthroughs (M_high_ = 5.45, M_low_ = 3.37, *p* < 0.001; M_high_ = 5.39, M_low_ = 3.46, *p* < 0.001). Thus, this experiment verified the validity of the video playback manipulation.

Finally, in order to exclude subjects’ pre-existing perceptions of environmental videos that were stimulated by different environmental short video adwords, all virtual video producers were used in this study; subjects were tested on video producer awareness, professionalism, credibility, and liking, with each dimension including three to four question items [14]. We also found no significant differences in subjects’ perceptions of virtual brand awareness, professionalism, credibility, and likeability (M_narrative video awareness_ = 4.11, M_non-narrative video awareness_ = 4.02, *p* = 0.46; M_narrative video professionalism_ = 4.02, M_non-narrative video professionalism_ = 4.12, *p* = 0.63; M_narrative video credibility_ = 4.19, M_non-narrative video credibility_ = 4.14, *p* = 0.45; M_narrative video liking_ = 2.91, M_non-narrative video liking_ = 4.01, *p* = 0.45). Hence, the stimulus materials selected for the pre-experiment could be used in the subsequent formal experiment [67]. After the pre-experiment, we discussed the video content and questionnaire, and the authors modified the short video and questionnaire based on the results of the discussion, resulting in the final experimental materials and questionnaire [68].

### 3.2. Formal Experiments

This study used an experimental method to test the above hypotheses, using message type (non-narrative vs. narrative), social media metrics (low vs. high play) and four experimental groups to conduct the experiment. This study used a questionnaire to collect data. The questionnaire was administered using a 7-point Likert scale to internet users who had previously used social media. The questionnaire was distributed through the online survey platform Questionnaire Star. Users were invited to participate in the survey through instant messaging software. A total of 356 questionnaires were collected and 61 invalid questionnaires were excluded, resulting in a valid questionnaire return rate of 82.86%. The exclusion criteria were the following: (1) people who had not used social media; (2) users who responded with 4 (don’t know) for all questions; (3) users who took less than 60 s to answer all of the questions. The experiment lasted 4–5 min, including the 3 min of video playback.

The final sample of 295 cases was included in the data analysis. The majority of the subjects were distributed between the ages of 20 and 49 (N = 229, 78%); their education levels were mostly bachelor’s (N = 172, 58.2%) and master’s (N = 86, 29.2%) degrees. The basic information of the sample is shown in Table 1.

### 3.3. Experimental Stimuli

We selected four short videos on the topic of environmental willingness (see 3.1 Pre-experiment section for details of the validation process), set as (1) non-narrative message, low play; (2) non-narrative message, high play; (3) narrative message, low play; and (4) narrative message, high play. The non-narrative video depicts the penalties faced by illegal fishing during the fishing ban on the Yangtze River in China [69]. The narrative video, on the other hand, portrays ShakeY in fishermen retiring from fishing and switching to production on the Yangtze River, highlighting the plot and timeline, and telling the story of a fisherman, using first-person, who retired from fishing and switched to production under the national policy [35]. Both videos are kept to 3 min in length, and both present the importance of natural conservation to the development of human society. Video play count is measured by four proxy variables (likes, retweets, favorites, comments) which appear at the bottom right of the short video [7]. The number of likes, retweets, favorites, and comments are considered high if any such parameter is above 10,000, and low if a parameter is below 200 [42].

### 3.4. Measurement

The questions in this study were measured using a 7-point Likert scale, with 1 indicating “strongly disagree” and 7 indicating “strongly agree”. In order to ensure the reliability and validity of each of the items, all scales were derived from established scales in published academic papers from overseas, and were translated in both directions in order to ensure the accuracy of the items in the questionnaire. Prior to the formal survey, 20 subjects were invited to take a pretest, in order to confirm the validity of the questionnaire [48]. Based on the feedback from the pretest, the experimental process was optimized to determine the final questionnaire items.

(1) The level of immersion scale developed by Risselada, H. et al. (2018) was borrowed and modified as appropriate, and three items were used to measure the subjects’ immersion level [10]. The items included “I forgot everything around me”, “I enthusiastically participated in this activity”, and “I was immersed in this activity” (M = 4.82, SD = 1.22, Cronbach’s α = 0.787).

(2) The measurement of social influence was based on the definition and measurement from Yang, J. et al. (2018) of social influence [11], using the following: “Sometimes people around me can give me the right decision”, “The views of my family and friends are important to me before I make a choice”, and “As far as society as a whole is concerned, most people behave in a way that is worthy of recognition” (M = 4.69, SD = 1.38, Cronbach’s α = 0.812).

(3) The measurement of “willingness to protect the environment” was based on Li Z G et al. (2021) and their willingness to protect the environment questionnaire [5], which ultimately used five statements to measure, as follows: “I am willing to participate in environmentally friendly public welfare activities”, “I prefer to choose environmentally friendly products and services”, “I am willing to reduce my own environmentally unfriendly behavior”, “I am willing to share environmental ideas with my friends around me”, and “I will choose an environmentally friendly lifestyle” (M = 4.42, SD = 1.23, Cronbach’s α = 0.783).

(4) Control variables included demographic characteristics and personality traits [12]. Personality traits were measured in terms of both empathy and image-generating ability (Laer et al., 2014): “I am very emotional”, “I can empathise with others”, “I am imaginative “, and “I am good at making associations” (M =4.17, SD = 0.88, Cronbach’s α = 0.809).

The overall reliability coefficient of the questionnaire in this study was tested using Cronbach’s alpha coefficient, which ranges from 0 to 1. In general, a Cronbach’s alpha coefficient greater than 0.7 is considered high reliability; below 0.35, it is considered low reliability; 0.5 is the minimum acceptable level of reliability [11]. The Cronbach’s alpha coefficients for immersion experience, social influence, willingness to protect the environment, and personality traits among the control variables were 0.787, 0.812, 0.783, and 0.809, respectively, all of which were greater than 0.7, indicating that the overall reliability of the questionnaire was high.

### 3.5. Data Analysis

This study used SPSS 25.0 (IBM New York, NY, USA) for data analysis, stepwise randomness tests, experimental stimulus manipulation tests, and independent sample *t*-tests, as well as ANOVAs and linear regression analyses based on the research questions and hypotheses. The correlation results for each continuous variable are shown in Table 2.

From Table 2 it can be seen that the correlations between subjects’ personality traits (control variables), immersion experience, and social influence, and individuals’ willingness to protect the environment, are significant. Therefore, we followed up with a regression analysis to further explore the causal relationship between these variables and willingness to protect the environment.

#### 3.5.1. Randomness Test

The 295 samples were allocated into four groups, with small differences in the numbers of subjects between the groups. A one-way ANOVA test was conducted on age, education, income, and personality traits. The results revealed that the differences between samples were not significant (*p* > 0.05), indicating that the subjects were randomly assigned to each experimental group. The results of the data analysis are shown in Table 3.

As can be seen in Table 3, the control variables (age, education level, income, and personality traits of the subjects) did not differ significantly after grouping, and the effect of the control variables on environmental willingness behavior between groups can be excluded.

#### 3.5.2. Experimental Stimulus Manipulation Test

The questionnaire used 3 statements to measure subjects’ assessments regarding non-narrative versus narrative videos: “I think the video is telling a story”, “I think the video has a plot”, and “I think the video has a timeline” [34]. One statement was used to measure subjects’ assessment regarding low versus high airplay: “I feel that the video has high airplay” [8]. Subjects responded using a 7-point Likert scale, with 1 to 7 referring to different, increasing levels of conformity. The results are shown in Table 4.

As shown in Table 4, the differences between subgroups are significant. The results of the independent samples *t*-test showed that both experimental stimuli were valid (*p* < 0.001).

#### 3.5.3. Research Questions and Analysis of Research Hypotheses

In order to test the experimental hypotheses, this study used a multivariate ANOVA to compare the roles of video type (non-narrative vs. narrative) and social media metrics (low vs. high play volume) on environmental intentions, immersion levels, and social influence. The distribution of key variables between groups is reported in Table 5.

As can be seen in Table 5, viewers had higher levels of immersive experience in the narrative type of short video subgroup; viewers had higher levels of perceived social impact in the non-narrative type of short video subgroup. Viewers’ willingness to protect the environment was higher in the subgroups with higher viewership.

## 4. Hypothesis Testing

### 4.1. Effect of Experimental Stimuli

In order to verify H1, the type of short video has an impact on the willingness to protect the environment, an independent-samples *t*-test was conducted. The results showed that there is no significant difference in the willingness to protect the environment between viewers of narrative vs. non-narrative types of short videos (M: 4.27 vs. 4.41, respectively; t = 1.48, *p* > 0.05); hence, Hypothesis 1 held.

In order to verify H2, watching narrative videos produces higher levels of immersion experience compared to non-narrative videos, an independent-samples t-test was conducted. The results showed that viewing narrative videos had a more positive effect on subjects’ immersion levels than non-narrative videos (M: 5.12 vs. 4.46, respectively; t = −2.75, *p* < 0.01); hence, Hypothesis 2 held.

In order to test H4, viewing non-narrative videos produces a higher level of social influence compared to narrative videos, an independent samples *t*-test was conducted. The results showed that subjects who watched non-narrative videos produced higher levels of perceived social influence compared to those who watched narrative videos (M: 5.32 vs. 4.59, respectively; t = 2.15, *p* < 0.01); hence, Hypothesis 4 held.

In order to test H6, watching a high-volume video produces a stronger immersion experience compared to a low-volume video, an independent samples *t*-test was conducted. The results showed that viewing videos with high-play volume produced a stronger immersion experience than viewing videos with low-play volume (M: 5.21 vs. 4.42, respectively; t = 1.23, *p* < 0.05); hence, Hypothesis 6 held.

In order to test H7, viewing high volume videos produces higher levels of social influence compared to low volume videos, an independent samples *t*-test was conducted. The results showed that viewing videos with high airplay produced higher levels of social influence than viewing videos with low airplay (M: 5.16 vs. 4.47, respectively; t = 1.19, *p* < 0.05); hence, Hypothesis 7 held. The results of the independent samples t-tests are shown in Table 6.

### 4.2. Linear Regression Analysis

In this study, linear regression analysis was conducted, with video type (non-narrative vs. narrative video) and social media indicators (low vs. high play) as the independent variables, while immersion experience, social influence, and environmental intention were the dependent variables. The results found that the type of message significantly influenced the level of immersion (β = 0.21, *p* < 0.01) and social influence (β = −0.16, *p* < 0.05) of the subjects. On the one hand, subjects who watched narrative videos produced higher levels of immersion experiences compared to non-narrative videos. On the other hand, subjects who watched non-narrative videos produced higher levels of social influence.

In order to understand the effects of the immersion experience and social influence on environmental intention, this study conducted a linear regression analysis using these two variables as independent variables, and environmental intention as the dependent variable. The data showed that the immersion experience was significantly and positively correlated with willingness to protect the environment (β = 0.34, *p* < 0.001), and that social influence was significantly and positively correlated with willingness to protect the environment (β = 0.28, *p* < 0.001). Hypothesis 3 and Hypothesis 5 held, meaning that the higher the subject’s immersion experience, the stronger their willingness to protect the environment; additionally, the higher the subjects’ perceived social influence, the stronger their willingness to protect the environment. The results of the linear regression analysis are shown in Table 7.

## 5. Discussion

The type of video (non-narrative vs. narrative) used in the environmental social media had no direct effect on people’s intention to protect the environment; however, their indirect effect was significant. This finding of no direct effect has been corroborated in other studies, where neither non-narrative nor narrative video had a significant direct effect on individuals’ attitudes toward refusing to use e-cigarettes [70]. The different findings on the direct effects of message type on attitudes may lie in differences in environmental behaviors, which researchers have suggested could mediate the role of message type [71]. Attitudes about some topics may be more firmly formed and less likely to be changed than others, and different topics may lead to different findings. Therefore, environmental protection as a topic needs more public participation; moreover, how to use social media to stimulate public participation is a topic worthy of in-depth study.

### 5.1. Theoretical Implications

(1)Enriching the study of immersion theory

In this study, it was found that the indirect effects of the type of information are possible through the immersion experience. That is, subjects who watched narrative videos reported higher levels of immersion experiences, and immersion experiences were significantly and positively correlated with environmental intentions; consistent with immersion experience theory, the processing of narrative information is based on immersion experiences, where individuals are immersed in the content of the information, and the ability to refute story claims is reduced through immersion experiences, making the narrative experience seem more like a real experience, based on emotional infection that causes attitude change (Green and Brock, 2000). In our everyday lives, we communicate with people through storytelling, which is a fundamental mode of human interaction; narrative is the basis for our understanding of the world around us. It is these unique abilities of narrative that make it a common and effective form of persuasion in communication practice.

(2)Extending the study of social influence theory

In this study, it was found that the indirect influence of message type can be achieved through social influence, i.e., subjects who watched non-narrative videos had higher levels of social influence, and social influence was significantly and positively correlated with environmental intentions. In this study, viewing non-narrative videos had a more positive effect on the subjects’ perception of social influence. The perceived social influence was the subjects’ perception of the prevalence of environmental willingness, and about the level of social acceptance of the environmental willingness approach. The scientific facts contained in the non-narrative videos are based on conclusions drawn from extensive research practice, and such results can be seen in a large number of people engaged in the same behaviors; subjects may interpret this as a normative cue, and hence generate a higher level of perceived social influence. Thus, a higher level of social influence can be generated through non-narrative forms of presentation (e.g., interpretation of laws and regulations) during the messaging process, which in turn can influence the subsequent behaviors of users.

### 5.2. Management Implications

When companies need to promote themselves through short videos (e.g., brand communication), the way in which the content is presented can affect the effectiveness of the promotion. If a narrative approach is used, it is easier to stimulate an immersive experience for the viewer, which subsequently resonates deeply within them. A non-narrative approach, on the other hand, stimulates the viewer’s perception of social influence, i.e., the attitudes of other viewers act as a code of conduct for the recipient of the video. When choosing how to present a video, we also need to pay attention to the combination of content and presentation; for example, videos with a storyline are more effective when presented in a narrative way, while videos with explanations, such as interpretations of laws and regulations as well as presentations of statistics, should be presented in a direct non-narrative way. In addition, watching videos with high viewership had a more positive effect on subjects’ willingness to protect the environment than watching videos with low viewership. Interaction data from real social media platforms also includes likes, retweets, comments, favorites, and hotness rankings, all of which can reflect the public’s approval of the quality and content of the video.

## 6. Conclusions

This study had many shortcomings. Firstly, the experimental sample was not representative [72]. The study used a convenience sample, recruited through WeChat friend circle postings and friend referrals, which resulted in a concentrated sample in terms of education and age, and may have resulted in a more homogeneous perception. Although the aim of this study was to understand the effects of social media videos on subjects’ environmental intentions, the findings were limited to a specific sample group. Secondly, the experimental material needed to be refined. Due to the limitations of editing techniques and video material, the experimental videos may have had problems, such as incoherent plot starts and ends as well as unclear message delivery, which may have led to poor dissemination of the experimental materials and inaccurate measurement. Finally, the questionnaire was not sufficiently complete, lacked pretesting of existing attitudes and knowledge of environmental willingness approaches, and neglected to assess the credibility and authenticity of the information.

There were two indirect paths of influence of message type on attitudes: immersion experience and social influence. The direct influence of media messages was insignificant, suggesting that research on the effects of environmental messages on communication needs to explore its mechanisms of action in greater depth, in order to uncover which influencing factors really have an effect. It is important to explore effective strategies to advocate for the adoption of environmentally friendly intentions by subjects, and to focus on audience profiling as well as clarifying the purpose of environmental concept communication when communicating with target groups.

## Figures and Tables

**Figure 1 ijerph-19-09520-f001:**
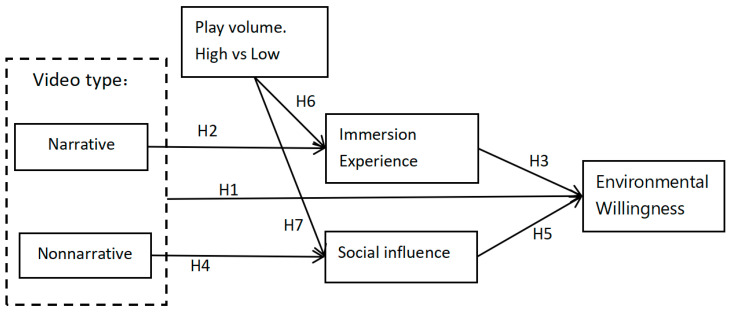
Hypothetical model for this study.

**Table 1 ijerph-19-09520-t001:** Basic information of the sample.

Attributes	Value	Frequency	Percentage (%)
**Sex**	Male	168	57
	Female	127	43
**age**	<19	59	20
	20–29	128	43.4
	30–39	87	29.5
	40–49	14	4.7
	50–59	7	2.4
**Education level**	High School and below	17	5.8
	Specialties	20	6.8
	Undergraduate	172	58.2
	Postgraduate	86	29.2
**Monthly income**	<2000 rmb	99	33.6
	2000–5000 rmb	85	28.8
	5000–10,000 rmb	65	22
	10,000–20,000 rmb	32	10.8
	>20,000 rmb	14	4.7

**Table 2 ijerph-19-09520-t002:** The correlation results for each continuous variable.

Variables	Personality Traits	Immersion Experience	Social Influence	Environmental Willingness
Personality Traits	1			
Immersion Experience	0.039	1		
Social Influence	0.071	0.262 **	1	
Environmental Willingness	0.132 *	0.277 **	0.469 **	1

Note: * *p* < 0.05; ** *p* < 0.01.

**Table 3 ijerph-19-09520-t003:** Random assignment validity check.

	Non-Narrative		Narrative		F-Value
	Low Play Count N = 72	High Play Count N = 73	Low Play Count N = 76	High Play Count N = 74	
**Age**	1.32 (0.51)	1.35 (0.45)	1.33 (0.52)	1.25 (0.45)	0.01 (*p* = 0.99)
**Education level**	2.37 (0.63)	2.45 (0.51)	2.51 (0.59)	2.47 (0.60)	0.74 (*p* = 0.16)
**Income**	2.14 (0.97)	2.08 (1.11)	2.34 (1.26)	2.07 (0.94)	0.77 (*p* = 0.55)
**Personality Traits**	5.68 (0.76)	5.61 (0.95)	5.46 (.96)	5.76 (0.72)	0.75 (*p* = 0.71)

**Table 4 ijerph-19-09520-t004:** Experimental stimulus manipulation test results.

		Frequency	Mean	St	T-Value
**Narrative** **vs.** **Non-** **narrative**	**Non-** **narrative**	145	3.89	1.16	−11.12 **
	**Narrative**	150	5.46	1.02	
**Play Volume** **High** **vs. Low**	**High Play Volume**	147	5.57	1.38	−12.41 **
	**Low** **Play Volume**	148	3.68	1.27	

Note: ** *p* < 0.01.

**Table 5 ijerph-19-09520-t005:** Inter-group distribution of continuous variables.

	Narrative		Non-narrative	
	High-Play Volume	Low-Play Volume	High-Play Volume	Low-Play Volume
**Immersion Experience**	5.12 (1.52)	4.66 (1.32)	4.51 (1.62)	4.42 (1.57)
**Social Influence**	4.77 (1.21)	4.79 (1.12)	5.12 (1.12)	5.13 (1.11)
**Environmental Willingness**	4.57 (1.46)	4.11 (1.30)	4.82 (1.28)	4.22 (1.12)

**Table 6 ijerph-19-09520-t006:** Results of the independent samples t-tests for the research hypotheses.

	Narrative vs. Non-Narrative		Play Volume		*t*-Test	
	Narrative	Non-Narrative	High	Low	Narrative vs. Non-Narrative	Play Volume
**Immersion Experience**	5.12 (1.52)	4.46 (1.32)	5.21 (1.62)	4.42 (1.57)	−2.75 **	1.23 *
**Social Influence**	4.59 (1.11)	5.32 (1.12)	5.16 (1.15)	4.47 (1.08)	2.15 *	1.19 *
**Environmental Willingness**	4.27 (1.41)	4.41 (1.30)	4.57 (1.37)	4.18 (1.19)	1.48	1.12 *

Note: * *p* < 0.05; ** *p* < 0.01.

**Table 7 ijerph-19-09520-t007:** Results of linear regression analysis.

	Immersion Experience	Social Influence	Environmental Willingness
**Age**	0.09	−0.03	0.01
**Education Level**	0.12	0.13	0.19 *
**Income**	−0.04	−0.02	0.10
**Personality Traits**	0.05	0.04	0.13 **
**Adjusted R2**	0.05	0.06	0.08
**Narrative** **vs. Non-** **narrative**	0.21 **	−0.16 *	−0.09
**Play Volume**	0.12 **	0.21 **	0.15 *
**Adjusted R2**	0.12	0.14	0.13 ***
**Immersion Experience**	--	--	0.34 ***
**Social Influence**	--	--	0.28 ***
**Adjusted R2**	--	--	0.37 ***

Note: * *p* < 0.05; ** *p* < 0.01; *** *p* < 0.001.

## Data Availability

Not applicable.

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
