# Peer review of "The Effects of the Type of Information Played in Environmentally Themed Short Videos on Social Media on People’s Willingness to Protect the Environment"

_ijerph, 2022, doi:10.3390/ijerph19159520_

Round 1

Reviewer 1 Report

The opening paragraph of the introduction needs to be better supported with references. 

Lines 69-73 do not clearly say what you intend to say. Please revise. 

Lines 76-81, again should be better supported with references.  The remainder of the section is well referenced. 

Line 157-168, reference(s)?

Line 354, perhaps you should delete the word "mainly".  Unless data was also collected in another way?

Overall, the foundation upon which this article is built should be better supported with citations of prior related work.  There are numerous statements that are not self evident and require references.  I have pointed out some.  Please check carefully for others. 

Section 3.1 Sample does not adequately describe the process of data acquisition.  More detail is needed. 

Section 3.2 More information supporting the justification for how the videos were developed is needed.  There must be some related research you can call upon to support your decisions here.  Length for example. Section 3.3 for example is well supported by citing related work. 

Discussion lacks implications and description of contributions. The results are clearly explained, however, the implications, or contribution to the existing body of knowledge is not evident. This ties into the comment above regarding the foundational work done previsously. Clearly explain both the theoretical and practical contributions of this work in building upon the prior work cited in the paper.  

The English and writing quality is overall quite high but there are a few minor typos throughout, check word spacing in particular.

Citability can be improved by giving a little more thought to the keywords. Check MDPI guide for this.  

Overall the article is well organized, the methodology appears sound, and while I did not check the statistics the findings are well supported by the data. 

If I could offer some advice of a proceedural nature. By the time you have added the necessary citations to support this paper, and format them for MDPI, and convert to MDPI numbering..... well, it will take many hours.  Consider using Zotero with the MDPI plugin to add and format your references.  You will save many hours. 

Reviewer 2 Report

The manuscript is clear concerning the main objetive and the contributions are relevant in the área os the impact of social media in the effects on People's Willingness Protect the Environment. The manuscript explore the effects of message type nonnarrative vs. narrative information, and two and social media metrics. The author point out some limitations to the research, however this  research can contribute to future investigations.

The literature review is well structured, and focus on the main studies on the reasearc areas. The literatura review also supported the 6 hypotheses to the present research.

In the methodology should explain who the participants are and when the data collection took place, and a subchapter with the data analysis.

The research is interesting but should have the implications in terms of research in the areas addressed, in terms of methodology and eventually in terms of the government, considering that the research addresses China's agreements with the United Nations General Assembly

There are some parts that need to be reviewed namely:

Figure 1. hypothesis model for this study:. could be improved to make it clearer how it intends to demonstrate the relationship between the variables and hypotheses, for example by putting each variable under analysis in separate text boxes.

Line 63 review “verorsus”

Line 157 author Csikszentmihalyi no date and not included in the references

Line 231: Social influence theory suggests that the influenced are influenced by the influencer in two main ways… review this sentence

Line 239 – behaviors substitute to behaviour, decide if it is to use American ou English.

Review the references:  authors are missing

Author Response

Dear reviewer:

Reviewer 3 Report

The topic is interesting and relevant.

The authors explain it in a clear way.

Along the paper authors state that no direct effect was found between type of messages and intention to protect the environment. Was that tested? This is not stated in the model and relates to no hipothesis.

Cronbach alpha of 'Level of immersion' was not indicated. Was it at the required levels? Authors should indicate it and explain what they measure and if they are at adequte levels.

All the tables at 3.4 were not anallyzed nor discussed. Authors refer to them but don't add any kind of interpretation. Reader's must know what information did they take from them.

4.2. Linear regression analysis. Data from the table does not match data from the text. What is included on the table, the betas? Again analysis and consistency between the text and tables must ne included.

Table information is also mandatory. Authors cannot assume that the readers know what * or ** means. It must be indicated.

For example on table 6, it must be indicated what the values mean. 

Finally, the references are incomplete. There are too many references along the text that are not indicated in the 'references' section.

Author Response

Dear reviewer:

Reviewer 4 Report

The reviewed article is about influencing audiences through social media videos on environmental issues. In it, the authors decided to look at, how narrative and non-narrative video materials influence viewers on an environmental issue. The authors formulated 3 main objectives, aiming to find out:

1) Are viewers better influenced persuasively by narrative or non-narrative materials?

2) Is the quantitative indicator of positively differentiating video materials for low or high viewership?

3) How do different types of persuasiveness affect the audience: formal and informational?

In pursuit of the 3 main goals, the authors formulated 6 hypotheses and conducted research on a sample of 295 cases, using mixed research methods, including observation and questionnaire, and using video as a tool in the study. In a cognitively valuable and needed study some doubts are raised by the reviewer not about the methodology in general, but about the specific diagnostic tool, namely the videos. This is particularly true of the format of these videos, i.e. documentary film. If we take the title of the study as a starting point , then we must conclude, that the documentary format is a creation perceived by social media audiences as non organic format, i.e. one, that does not fit into the social communication environment and is therefore not treated by audiences as a source of information, which they then integrate willingly into their own system of perception of the surrounding world. Social media audiences around the world, including those in the group of interest to the researchers, rely on influencer recommendations rather than documentaries. The decision to choose the format therefore leaves us with laboratory conditions for evoking social engagement rather than natural conditions, as in everyday using of social media. We therefore do not know how audiences would have reacted, if the videos had been prepared based on the dominant practices of video expression in social media. A significant methodological weakness, which is also pointed out by the authors of the study, is also questionable and deserves a negative evaluation. This is the allegation of the unrepresentativeness of the sample recruited from the circle of acquaintances therefore of people with demographic characteristics similar to those of the authors of the study. Taking into account the fact, that the authors themselves pointed out the potential weaknessesof the research procedure they designed, I propose, that the reviewer's objection about the legitimacy of using the documentary film format colud be accompanied by a justification, precisely showing the authors' point.

Author Response

Dear reviewer

Round 2

Reviewer 1 Report

Revisions are exactly as requested. 

Congratulations on a fine piece of work.  I believe it should be published in its current form and I wish you all the best. 

Reviewer 2 Report

The authors have improved the introduction and responded to all the suggestions made.

The concepts and theories are better grounded and supported.

They have greatly improved the methodology part and the explanation and presentation of the main results.

The Management implications were introduced, but could have been adjusted to the context of the research developed.

Reviewer 3 Report

Please review the references.